# In Vivo Digestibility and In Vitro Fermentation of High Dietary Fiber Forages in Growing Pigs' Diets

Mónica Gandarillas [1], María Isidora Valenzuela [2], Jorge Molina [2], Rodrigo Arias [1] and Juan Keim [1,*]

1   Institute of Animal Production, Faculty of Agricultural and Food Sciences, Universidad Austral de Chile, Valdivia 5110566, Chile; monica.gandarillas@uach.cl (M.G.); rodrigo.arias@uach.cl (R.A.)
2   Graduate School, Faculty of Agricultural and Food Sciences, Universidad Austral de Chile, Valdivia 5110566, Chile; isi.valenzuelarios@gmail.com (M.I.V.); jorge.a.molina1994@gmail.com (J.M.)
*   Correspondence: juan.keim@uach.cl; Tel.: +56-63-2293659

**Abstract:** The pig farming industry is constantly challenged to seek low-cost ingredients that fulfill animal requirements. In this study, two summer forage brassica meals were assessed as sources of dietary fiber in growing pigs by in vivo digestibility and in vitro fermentation experiments. The control diet included corn, soybean meal, and wheat middlings. The experimental diets replaced wheat middlings (15%) with turnip (*Brassica rapa*) roots or fodder rape (*Brassica napus*) whole plant meal, respectively. All diets were elaborated to be iso-nitrogenous and iso-energetic. The turnip diet had a greater digestibility rate for gross energy ($p = 0.020$). The ash digestibility was greater for the rape diet and intermediate for the turnip diet, with the lowest value for the control diet ($p = 0.003$). When incubating pure brassica forages, only gas production at 72 h was greater for the turnip than rape diet ($p = 0.04$). No differences ($p > 0.05$) in the in vitro gas production parameters were observed among the diets. The pure fermentation of turnip increased the VFA concentration and propionate molar proportion, whereas acetate was reduced ($p < 0.05$), which resulted in a trend towards a greater propionate molar proportion with the inclusion of turnip in the diet ($p = 0.067$). The inclusion of 15% of turnip meal increased the in vivo energy digestibility and tended to modify the fermentation parameters, increasing the molar proportion of propionate, whereas the inclusion of whole plant fodder rape did not affect the in vivo digestibility or in vitro fermentation compared with the control diet.

**Keywords:** fermentation; rape; turnip

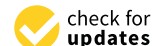



## 1. Introduction

Pig nutritionists are constantly challenged to seek new ingredients to reduce feedstuff costs by replacing grains that are also consumable for humans [1]. In sows and boars, high-fiber diets may benefit body condition and gastric fill without affecting reproductive traits [2]. In contrast, diets for growing pigs traditionally do not include high levels of dietary fiber because fiber decreases the digestible energy from the diet and some amino acids, limiting their potential growth [3,4]. However, due to the potential benefits to gut health, the immune system, reproductive traits, and welfare, dietary fiber has been evaluated as a partial replacement for cereals [5,6].

Summer brassicas are annual crops that are generally used as supplements for ruminants in times of seasonal shortage [7,8], including turnips (*Brassica rapa* L. subsp. *rapa*) and fodder rape (*B. napus* L. subsp. *biennis*) [7]. Both are high in dietary fiber (DF), with a chemical composition that varies as the leaf/bulb–stem ratio fluctuates [9]. Brassica crops contain insoluble fiber (IF) in terms of NDF ranging from 18–24% for turnip and rape, whereas the soluble fiber (SF) ranges from 24–38% [9]. The SF is mainly composed of pectins (7–9%), galactans, and β-glucans, among others [7,8,10,11]. Although research focused on summer brassica inclusion in pig diets is limited, it has been shown that forage

rape and turnips can be included in the diet without negative repercussions in terms of animals' oral perception [12].

Pigs are able to digest DF in their hindgut and absorb microbial fermentation end-products such as short-chain fatty acids, chain fatty acids, lactate, and amines [6]. No negative effects on pig performance with the inclusion of DF (ranging between 19 and 30%) from different byproducts has been reported [13]; moreover, fiber contributes to gut health, including fermentation characteristics and immune responses [14]. Its inclusion level is dependent on many factors such as the animal's age [15] and physiological state [16] and the characteristics of the feed ingredient [17].

The nutritional value of a food or ration can be expressed through the digestibility coefficient. However, the digestible proportion of a nutrient can be modified by the bacteria of the large intestine, changing the amino acid profile of the ileal content with respect to the profile in the feces [18]. Thus, post-gastric fermentation is relevant. The in vitro gas production technique has been used to describe fermentation in the rumen [19]. However, it can be adapted to assess the potential fermentability of feed ingredients for monogastric diets [20]. Although brassica forages have been extensively studied in ruminants, their use as a DF ingredient in pig diets lacks information. Recently, a pilot study showed that forage brassicas can replace wheat middlings in the diet without affecting the pigs' acceptability or palatability [12]; however, to the best of our knowledge, no information regarding its digestibility and in vitro post-gastric fermentation has been reported. This information would help in increasing the gap of knowledge about the inclusion of new high-fiber ingredients in pigs' diets. Thus, the objective of this study was to assess the in vivo digestibility and in vitro fermentation of two meals elaborated from summer brassicas as a replacement for wheat middlings in the growing pig's diet.

## 2. Materials and Methods

### 2.1. Brassica Meal Elaboration and Animal Housing

Turnips (4 kg seed/ha) and forage rape (5 kg/ha) were established in October 2017 were grown at Universidad Austral de Chile's Agricultural Research Station (39°47′ S, 73°13′ W) in Typic Hapludand soil. Brassicas were harvested in the morning between 05:00 and 07:00 at 100 days after plant emergence. All plants were in a vegetative stage of growth at harvest. Summer turnips were collected manually, with leaves separated from the bulbs and soil attached to the bulbs being removed. Rape plants were harvested with a cutter-bar mower (model 140; Bertolini, Reggio Emilia, Italy) at 5 cm above ground level. Plants of fodder rape and roots of summer turnip were dried in an oven with air-forced ventilation at 60 °C for 72 h or until reaching a constant weight. After drying, the plant materials were milled to 5 mm and stored in double bags and labeled for later use.

Experiments were carried out from December 2018 to January 2019. Six healthy castrated male pigs (PB 337 × Camborough 22, PIC genetics), at 25.2 ± 1.1 kg of live weight and 70 days old. were randomly allocated into three dietary treatments (Control, Turnip, Rape) in a replicated 3 × 3 Latin square design. The three diets were formulated as described in Table 1. Previously, all feedstuffs were chemically analyzed for their dry matter (DM), crude fiber (CF), ether extract, ash, and acid detergent fiber (ADF) contents according to [21] (procedures 978.10, 942.05, 920.39, and 973.18, respectively). The gross energy was assessed using adiabatic calorimetry [21] and the NDF content using a heat-stable amylase [22]. The nitrogen content was measured by combustion (Model FP-428 Nitrogen Determinator; LECO, St. Joseph, MI, USA) and was used to calculate the CP content (N × 6.25). All diets were carried out to be iso-nitrogenous and iso-energetic. All experimental diets were elaborated in an animal feed processor.

The pigs were allocated to a common pen (5.2 × 8.0 m) for 5 days. Before the onset of the experiment, the pigs were housed (5 days) in individual units (1.70 × 0.85 m) for adaptation to the experimental conditions. During this period, the pigs were fed with a commercial concentrate. The individual units had a concrete floor with ad libitum access to water through a stainless steel nipple. Sawdust was used as bedding material only during

the adaptation period. The light regime was 16:8 h of light and darkness, respectively. The temperature was set to 20 ± 4 °C and the relative humidity to 50–60% (RH), with artificial ventilation between 08:00 and 21:00.

**Table 1.** Ingredients and chemical compositions of the experimental diets. Brassica crop meal was included at 15% of the diet by replacing wheat middlings included in the control diet.

|  | **Control** | **Turnip** | **Rape** |
|---|---|---|---|
| Ingredients |  |  |  |
| Corn (%) | 42.1 | 42.1 | 42.1 |
| Soybean meal (%) | 33.5 | 33.5 | 33.5 |
| Soybean oil (%) | 5.5 | 5.5 | 5.5 |
| Wheat middlings (%) | 15 | 0 | 0 |
| Turnip meal (%) | 0 | 15 | 0 |
| Forage rape meal (%) | 0 | 0 | 15 |
| Calcium carbonate (%) | 0.4 | 0.4 | 0.4 |
| Salt (%) | 0.35 | 0.35 | 0.35 |
| Calcium biphosphate (%) | 1.15 | 1.15 | 1.15 |
| Premix vit-min (%) | 1.5 | 1.5 | 1.5 |
| Celite (%) | 0.5 | 0.5 | 0.5 |
| Overall (%) | 100 | 100 | 100 |
| Nutrient Concentration |  |  |  |
| Dry matter (%) | 88.4 | 89.29 | 89.23 |
| Ash (%) | 6.68 | 8.17 | 7.04 |
| Crude protein (%) | 20.93 | 20.91 | 20.58 |
| Ether extract (%) | 8.21 | 7.5 | 7.72 |
| Crude fiber (%) | 3.96 | 2.49 | 2.49 |
| NDF (%) | 13.16 | 10.46 | 10.72 |
| ADF (%) | 4.91 | 5.15 | 5.51 |
| Starch (%) [1] | 31.14 | 29.76 | 28.27 |
| Sugars (%) [1] | 4.38 | 7.67 | 3.97 |
| GE (kcal/kg) [1] | 4123 | 4122 | 4117 |
| DE (kcal/kg) [1] | 3442 | 3559 | 3543 |
| ME (kcal/kg) [1] | 3284 | 3401 | 3384 |
| NE (kcal/kg) [1] | 2502 | 2521 | 2511 |

[1] Values were predicted using Evapig® software [23]; NDF (neutral detergent fiber); ADF (acid detergent fiber); GE (gross energy); DE (digestible energy); ME (metabolizable energy); NE (net energy).

### 2.2. Experiment 1: In Vivo Digestibility

For experiment 1 (in vivo digestibility), the animals were fed ad libitum according to weight and age, and the maximum possible consumption was calculated for each pig. The feeders were refilled twice daily, at 09:00 and 17:00. The length of the experiment was 21 days, divided into three 7-day periods. In each 7-day period, a pair of pigs were randomly fed with one of the three diets. Each 7-day period was subdivided into 4 days of adaptation to the diet and 3 days of fecal collection, avoiding contamination. In those days, feeding was restricted to 85% of the maximum consumption. The pooled samples were stored in plastic bags, labeled, and frozen at −20 °C. A minimum of 75 g/pig/day of feces was collected. Prior to the analysis, the fecal samples were thawed for 2 days in a refrigerator at 3 °C. Daily samples for each treatment and pig were mixed and homogenized, obtaining an aliquot per treatment. The diets and feces were analyzed in triplicate, including the DM (in an air-forced oven at 60 °C for 48 h and then at 105 °C for 12 h [24,25]), organic matter (OM), CP, EE, gross energy (GE), NDF, and ADF contents. The total ash was obtained by calcining the sample in a muffle at 600 °C for 2 h (Thermolyne, Sybron Type 6000 Furnace) [21,24] and the acid-insoluble ash (AIA) was determined as an internal marker [25]. The apparent fecal digestibility was estimated using the indirect method through an indigestible marker [26]. As a marker, 0.5% celite (acid-indigestible ash-celite® 545 CAS 68855-54-9, Merck) was added to the diet to indirectly determine the

concentrations of nutrients and energy in the feces. The digestibility of nutrients and energy was determined by the equation used by Adeola [26]:

$$Fecal\ apparent\ digestibility\ (\%) = 100 \times \left(1 - \frac{diet\ marker}{feces\ marker} \times \frac{nutrient\ feces}{nutrient\ diet}\right)$$

*2.3. Experiment 2: In Vitro Fermentation*

Experiment 2 (in vitro fermentation) began with the preparation of the solutions to simulate gastric and small intestine digestion by using an in vitro enzymatic method [27]. The enzymatic solution preparation process started with dissolving 8.5 mL HCl in distilled water, making up to one liter, then 1.67 g of pepsin was subsequently added. A pancreatin buffer solution was prepared with 560 mL of disodium phosphate (0.165 mol/L) and 440 mL of dehydrated potassium phosphate at the same concentration. Then, 1.33 g of pancreatin, 53 mg of lipase, and 106 mg of bile salts were added. A third solution was prepared corresponding to HCl/acetic acid 1 mol/L each in a 5:2 ratio. Later, 50 g of each experimental diet sample was incubated. The enzymatic digestion was carried out in two liter beakers, with 850 mL of pepsin/HCl solution. The glass jars were placed in a water bath at 40 °C, with constant stirring for one hour. Subsequently, 225 mL of a sodium bicarbonate solution was added to achieve a concentration of 0.39 mol/L to be neutralized. At the same time, the pancreatin buffer solution was heated to 40 °C and then added to the neutralized pepsin/HCl solution. This mixture was kept at 40 °C with constant stirring. Then, 160 mL of the HCl/acetic acid solution was added to stop the enzymatic reaction. Finally, the solution was filtered with crucibles and the residue was dried in an air-forced oven at 60 °C until reaching constant weight to determine the dry matter content. The residues were used as substrates for the subsequent in vitro post-gastric fermentation test.

The post-gastric fermentability of the diets was evaluated through the in vitro gas production technique [28]. The evaluation of the in vitro gas production was made in three incubation runs, each corresponding to one of the three experimental periods (blocking factor). For each incubation, bottles with the three dietary treatments, the pure brassicas (turnip and rape), and blank bottles were used. The treatments were analyzed in triplicate. A total of 21 bottles (160 mL of volume) per incubation were added at 0.5 g/DM of each substrate and placed in a water bath at 39 °C. For each incubation, samples of each pig's feces (6.5 g) were mixed, forming a single pool that was diluted in 180 mL of 0.9% sodium chloride solution, being homogenized for one minute in a mixer machine. The fecal inoculum was prepared from feces collected from the 6 pigs housed in the monogastric unit at the research station. Feces collection was carried out 3 times within 21 days, taken directly from the rectum of each animal to avoid contamination. The feces were stored in plastic bags and then in a thermos previously filled with hot water at 60 °C, and then immediately transported to the laboratory. Later, each sample was leaked and added to a phosphate bicarbonate buffer solution, at a 1:16.8 weight/volume ratio [20]. The inoculum preparation was under a constant flow of $CO_2$ to maintain anaerobic conditions. Then, 89 mL of the buffered inoculum solution was applied to each bottle under a constant flow of $CO_2$. Finally, the initial gas pressure of each bottle was released through the extraction of the gas with a syringe connected through a three-way valve to a pressure transducer (PCE Instruments, Tobarra, Albacete, Spain), until the pressure reached 0.00 psi. Each bottle was then sealed and placed in a water bath at 39 °C for 72 h. The measurements of gas pressure (psi) and gas volume (mL) were recorded at 2, 4, 6, 8, 12, 18, 24, 48, and 72 h. The gas produced was released with a syringe until the pressure in the transducer read 0.00 psi. The fermentation was finished at 72 h by placing the bottles on ice. Each triplicate sample was subsequently taken to a beaker, where the samples were pooled. With a micropipette, 2 mL samples were collected and added to Eppendorf polypropylene microtubes (two per pooled sample). These samples were frozen at −20 °C for the subsequent analysis of VFA and $NH_3$.

The gas production data were adjusted to the model developed by France et al. [29]:

$$G = A\left(1 - \exp^{(-[b(t-L)+c(\sqrt{t}-\sqrt{L})])}\right)$$

where: G = the gas produced at a specific time (*t*); A = asymptotic gas production (ml/g DM); L = the lag time before the fermentation started (*h*); the constants b ($h^{-1}$) and c ($h^{-1/2}$) determine the fractional degradation rate of the substrate μ ($h^{-1}$), which is postulated to vary with time as proposed by Jang et al. [30]:

$$\mu = b + c/(2\sqrt{t}), \text{ if } t \geq L$$

The kinetics parameters of the gas production (A, $t_{1/2}$, G72, and μ at $t_{1/2}$) were compared in the statistical analysis, with $t_{1/2}$ representing the time to produce half of the asymptote volume.

The VFAs were analyzed through a GC-2010 gas chromatograph (GC) (Shimadzu Corporation, Kyoto, KYT, Japan) equipped with a GC Capillary Column (SGE, BP21 (FFAP), temperature range (C) = 35–240/250, UOM = EA) at the Institute of Food Science and Technology at the Universidad Austral de Chile. Samples in Eppendorf tubes (2 mL) obtained at the end of the fermentation phase were taken to the laboratory, thawed at room temperature, shaken, and centrifuged for 15 min at 12,000× *g*, then the supernatant was separated in 1.5 mL Eppendorf tubes for a later analysis. The VFA analysis was carried out as follows. In GC vials, 50 μL of hexanoic acid, 100 μL of formic acid, and 900 μL of the supernatant corresponding to each of the samples were added with a micropipette. The GC vials were passed through an AOC-20i auto-injector to determine the concentration (mmol/L) of VFA. Two solutions were used to determine the $NH_3$ content with a spectrophotometer (Spectroquant® Prove 600, Merck, Germany) following the protocol reported by [31]. The first solution (A) was composed of phenol and sodium nitroprusside and the second (B) with sodium hydroxide plus sodium hypochlorite. After 30 min, the absorbance was measured at a wavelength of 625 nm.

### 2.4. Statistical Analyses

Experiment 1: The pigs were randomly allocated into three dietary treatments in a replicated 3 × 3 Latin square design and analyzed using the MIXED procedure of SAS (9.4; SAS Inst. Inc.; Cary, NC, USA), considering the diet consumed (control, turnip, rape) as a fixed effect, the random effect of the square; the random effect of pig nested within the square, and the fixed effect of the period. The statistical model used was: $Y_{ik(j)l} = \mu + \alpha_{i(l)} + \tau_j + \beta_k + \delta_l + \varepsilon_{i(j)kl}$; where $Y_{ik(j)l}$ is the response variable of fecal apparent digestibility (%) and μ is the population mean; $\alpha_{i(l)}$ is the random effect of each pig (*i* = 1 to 6) within the square (*l* = 1 to 3); $\tau_j$ is the effect of the diet (*j* = 1 to 3); $\beta_k$ is the fixed effect of the period (*k* 1 to 3); $\delta_l$ is the random effect of the square (*l* = 1 to 3); $\varepsilon_{i(j)kl}$ is the experimental error associated with each observation.

Experiment 2: The data were analyzed as a randomized block complete design, with three blocks being the blocking factor, and the incubation run was performed with the following model: $Y_{ij} = \mu + \alpha_i + \beta_j + \varepsilon_{ij}$; where $Y_{ij}$ is the response variable, μ is the population mean; $\alpha_i$ is the effect of the diet (*i* = 1 to 3); $\beta_j$ is the random effect of the incubation run (*j* = 1 to 3); $\varepsilon_{ij}$ is the experimental error associated with the *ij*th observation. The data were analyzed separately for diets and pure ingredients. Each bottle was considered an analytical replicate. For both experiments, the mean values are presented as least square means with a significance of 5%. The least square means comparison between treatments was carried out using the adjusted Tukey test and PDIFF command in SAS.

## 3. Results

### 3.1. Experiment 1: In Vivo Digestibility

The apparent fecal digestibility levels of the nutrients and energy are shown in Table 2. There were differences in the apparent fecal digestibility of the gross energy (*p* = 0.02) and

ash ($p < 0.01$). The turnip diet had a greater digestibility for gross energy, followed by the control and rape diets, without differences between them. The ash digestibility was greater for the rape diet and intermediate for the turnip diet, whereas the lowest value was observed for the control diet. The dietary treatments did not affect the DM, OM, CP, EE, aNDFom, and ADFom digestibility ($p > 0.05$).

**Table 2.** Apparent fecal digestibility of the experimental diets of nutrients and energy in growing pigs.

| Nutrient | Fecal Apparent Digestibility (In Vivo) (%) [1] | | | *p*-Value |
| --- | --- | --- | --- | --- |
| | Control | Turnip | Rape | |
| Dry matter | 94.8 ± 1.0 | 93.4 ± 1.0 | 93.6 ± 1.3 | 0.594 |
| Organic matter | 97.6 ± 0.2 | 98.2 ± 0.2 | 97.4 ± 0.3 | 0.159 |
| Gross energy | 80.6 ± 1.1 [b] | 85.3 ± 1.1 [a] | 80.6 ± 1.4 [b] | 0.020 |
| Crude protein | 82.5 ± 1.0 | 81.4 ± 1.0 | 81.2 ± 1.3 | 0.501 |
| Ether extract | 81.5 ± 1.7 | 82.1 ± 1.7 | 79.2 ± 2.2 | 0.495 |
| Ash | 53.8 ± 2.0 [c] | 61.5 ± 2.0 [b] | 68.8 ± 2.6 [a] | 0.003 |
| aNDFom | 58.4 ± 3.1 | 59.1 ± 3.1 | 60.4 ± 4.1 | 0.904 |
| ADFom | 49.4 ± 3.2 | 58.4 ± 3.2 | 53.8 ± 4.2 | 0.152 |

[1] Values represent the least square mean ± standard error of the mean; aNDFom: neutral fiber detergent of the organic matter; ADFom: acid detergent fiber of the organic matter; different superscript letters within a row indicate statistical difference ($p < 0.05$) between diets.

### 3.2. Experiment 2: In Vitro Fermentation

There were no differences ($p > 0.05$) for any of the in vitro gas production parameters (GP72, A, A$1/2$, L, t$1/2$, and u among experimental diets; Table 3). When comparing pure brassicas, GP72 was 103 mL greater for turnip than rape ($p = 0.04$). However, the other parameters did not differ when comparing pure turnip roots and whole rape plants ($p > 0.05$).

**Table 3.** In vitro gas production parameters for diets and pure turnip and rape.

| | Diets | | | SEM | *p*-Value | Ingredients | | SEM | *p*-Value |
| --- | --- | --- | --- | --- | --- | --- | --- | --- | --- |
| | Control | Rape | Turnip | | | Turnip | Rape | | |
| GP72 | 213 | 193 | 201 | 5.4 | 0.209 | 267 [a] | 171 [b] | 19.9 | 0.040 |
| A | 238 | 207 | 238 | 11.5 | 0.184 | 262 | 221 | 19.2 | 0.345 |
| A$1/2$ | 119 | 104 | 119 | 5.8 | 0.184 | 131 | 111 | 9.6 | 0.345 |
| L | 1.37 | 2.46 | 2.35 | 0.5 | 0.453 | 1.4 | 1.73 | 0.8 | 0.482 |
| t$1/2$ | 29.4 | 30.4 | 26.6 | 4.2 | 0.851 | 15.6 | 29.0 | 5.2 | 0.285 |
| u | 0.07 | 0.09 | 0.04 | 0.02 | 0.503 | 0.11 | 0.04 | 0.02 | 0.601 |

GP72: Gas production at 72 h (mL/g DM); A: asymptotic gas production (mL/g DM); A$1/2$: gas production at t$1/2$; L (mL/g DM): Lag time (h); t$1/2$: time to ferment half of A (h); u: rate of gas production at t$1/2$. SEM: standard error of the mean. Different superscript letters within a row indicate statistical difference ($p < 0.05$) between diets or ingredients.

The ammonia, VFA concentration, acetate, and butyrate molar proportions were not affected by the replacement of wheat middling with rape or turnip ($p > 0.05$; Table 4), whereas the propionate molar proportion tended to increase with the inclusion of turnip in the diet ($p = 0.07$). When comparing the fermentation of pure brassicas, turnip increased the concentration of VFA and propionate and reduced the acetate content ($p < 0.05$), without differences for butyrate ($p > 0.05$).

**Table 4.** In vitro ammonia and VFA concentrations for diets and pure turnip and rape.

| | Diet | | | SEM | *p*-Value | Ingredients | | SEM | *p*-Value |
|---|---|---|---|---|---|---|---|---|---|
| | Control | Rape | Turnip | | | Rape | Turnip | | |
| NH$_3$ | 197.2 | 280.0 | 233.0 | 35.5 | 0.353 | 268.9 | 294.4 | 36.4 | 0.622 |
| VFA (mmol/L) | 41.6 | 43.5 | 44.2 | 2.1 | 0.719 | 46.2 [b] | 55.2 [a] | 1.0 | 0.01 |
| Relative proportion of total VFA (mmol 100 mmol$^{-1}$) | | | | | | | | | |
| Acetic | 44.4 | 46.6 | 47.1 | 1.9 | 0.690 | 60.7 [x] | 53.9 [y] | 1.5 | 0.077 |
| Propionic | 30.2 [y] | 30.5 [y] | 34.3 [x] | 1.0 | 0.067 | 24.4 [b] | 30.9 [a] | 1.4 | 0.045 |
| Butyric | 24.7 | 22.9 | 18.9 | 1.6 | 0.109 | 14.9 | 15.2 | 0.5 | 0.703 |

NH$_3$: ammonia; VFA: volatile fatty acids; [a,b] indicates statistical difference ($p < 0.05$) and [x,y] trend ($p < 0.1$) between diets or ingredients. SEM: standard error of the mean.

## 4. Discussion

It is worth mentioning that although the diets were iso-nitrogenous and iso-energetic, the energy sources among the diets were different, meaning the results found in this study may be related to these changes. Nevertheless, the main aim was to evaluate new high-DF ingredients.

Our in vivo digestibility results do not agree with previous studies evaluating high-DF feeds, where a high level of DF had a negative impact on the DM, OM, CP, EE, and energy digestibility [6,32]. The energy digestibility decreases as the DF increases, since nutrients are adsorbed to the fiber particles and further transported to the large intestine, where they are fermented and transformed into VFA [4], resulting is an inverse relationship between the fiber content and energy digestibility [33]. Moreover, it is well documented that the SF and IF digestibility coefficients are different due to their physical–chemical properties within the digestive tract of non-ruminant species such as pigs [34].

Diets with high levels of IF may decrease the intestinal transit time because IF limits the access and action of endogenous enzymes in the small intestine and microbial fermentation in the large intestine [1,35]. Although in this experiment the percentages of IF and SF were different among the diets, the lack of differences in digestibility among the brassica diets may be due to the low level of inclusion. A negative effect in terms of energy digestibility was reported when alfalfa meal was included at 20% in the diet [36] and for wheat middlings at 50% of the diet [37]. In another study, an inclusion level of 15% with three sources of fiber (beet pulp, soybean hulls, and pectin residue) resulted in a decreased digestibility of the energy, CP, EE, and non-soluble protein [38]. In our study, the energy digestibility of the turnip diet (high in SF) was greater than the control diet (high in IF) and the rape diet (high in SF). A possible explanation could be that the turnip meal was elaborated from the roots, which are high in sugars and starch, whereas the rape meal was elaborated from the whole plant, having a higher fiber content [9].

The lack of differences in CP digestibility among the diets in this experiment agrees with a previous study [39]. However, another study has shown that increasing the SF in the diet increases the nitrogen digestibility as compared with a high-IF diet [40]. Endogenous nitrogen and bacterial biomass losses constitute a loss of nitrogen, resulting in lower nitrogen digestibility values [40]. Additionally, it is known that the increase in bacterial mass is caused by an increase in SF [41], since SF acts as a prebiotic, thereby providing energy to the beneficial hindgut bacteria [42].

It has been reported that the inclusion of IF in pig diets has a greater effect on fat digestibility than SF [38,43]. However, in our study, we did not find effects on either extract's digestibility. The ash digestibility was greater in the rape diet as compared to the turnip diet, which in turn was greater than the control diet. The last factor can be explained, since the two brassica ingredients contained higher ash contents as they were harvested directly from the soil in the farm, in contrast to the wheat middlings from the control diet, which came from a milling plant.

The lack of differences in the NDF and ADF digestibility levels among diets may be because the SF is not accounted for either in the NDF or in the ADF fractions [44]. Additionally, it has been reported that in gestating sows diets, high quantities of SF could increase the SF and IF digestibility values when using sugar beet pulp as the source of fiber [40]. An additional benefit of ingredients high in SF is that they allow greater fermentation of the insoluble fraction, probably because SF increases the microbial population in the intestine [1,45].

It should be noted that in this study the feces were used as an inoculum, whereas others used bovine ruminal liquor [1]. Most of the abundantly identified taxa in the rumen corresponded to members of the genera *Prevotella*, *Ruminococcus*, and *Butyrivibrio*, as well as unclassified members of the orders *Clostridiales* and *Bacteroidales* and of the families *Ruminococcaceae* and *Lachnospiraceae* [46], whereas the main bacteria found in the hindgut of pigs are of *Streptococcus* spp., *Lactobacillus* spp., *Eubacterium* spp., *Fusobacterium* spp., *Bacteroides* spp., *Peptostreptococcus* spp., *Bifidobacterium* spp., *Selenomonas* spp., *Butyrivibrio* spp., *Prevotella* spp., and *Ruminococcus* spp. [1,45]. This could explain the differences in GP associated with microorganisms in the rumen of cows and the large intestine of pigs. A possible explanation for the higher GP72 h found in the turnip meal could be due to the use of only the roots of turnip plants to make the meal, instead of the whole plant as for the rape meal. The roots of turnip are richer in total soluble sugars (208–248 g of sugars/kg DM) and have slightly greater concentrations of starch (72–165 g/kg DM) compared with whole rape plants (138 and 98 g/kg DM for sugars and starch, respectively) [9]. The results for the GP are in accordance with the VFA concentrations, as there is a strong relationship between GP and VFA concentrations because the microbial fermentation of the digest in the cecum and colon in pigs produces VFAs, lactate, amines, indoles, and phenols, as well as various gases such as hydrogen, carbon dioxide, and methane [14,47]. Digestion in the large intestine takes the longest amount of time (20–38 h) [48], which is why the bacteria would digest and ferment carbohydrates and proteins in greater quantities, obtaining more fatty acids in both short- and mid-chain VFAs. Additionally, turnip contains a greater amount of sugar than wheat middlings and rape, potentially changing the fermentation patterns [12,39]. Previous in vitro studies carried out with human fecal bacteria have shown that the fermentation of starch produces a higher proportion of butyric acid as a product of short-chain VFAs when compared to the fermentation of foods rich in pectin, such as brassicas that contain more than 300 g/kg DM of SF [40]. It is probable that the resistant starch in the diet that escapes from the digestion of the small intestine is butyrogenic [49]. Starch fermentation increases the proportion of propionic and butyric acids, while propionic acid stimulates the absorption of water from colon digestion and butyric acid is the preferred energy source for colonocytes [32,50]. Brassicas have higher concentrations of easily fermentable carbohydrates, such as sugars [7,9], which would increase the amount of propionate, as observed with the inclusion of turnip compared with the other diets.

The trend of a higher concentration of propionic acid may be associated with its gluconeogenic or sugar-forming property, which converts pyruvate and three- and four-carbon compounds into glucose [2]. Likewise, the process of drying the meal could have had a positive effect on starch digestibility. In addition, the non-starch polysaccharides and lignin, which are forms of dietary fiber, must be considered. The main destination of these dietary fibers is the digestion and catabolism of the intestinal microflora to short-chain VFAs [1,45]. It has been reported that resistant starch produces an increase in butyric concentrations, gums increase propionic acid, and pectin increases acetic acid [51].

Based on our results, where no differences in terms of digestibility were found, and those by Figueroa et al. [12], who found that forage brassica meals did not negatively affect the preference or acceptability of the diet when incorporated at 15% in growing pig diets, we expect no differences in terms of growth performance. However, this needs to be evaluated in further research. The increase in the propionate molar proportion with the inclusion of summer turnip root meal could benefit pregnant sows, whose daily consumption is often restricted to small meals to avoid being overweight at farrowing.

## 5. Conclusions

The replacement of wheat middlings with 15% of turnip or rape meal in the diet of growing pigs increased the in vivo energy and ash digestibility, and the inclusion of turnip root tended to modify the in vitro fermentation parameters, increasing the molar proportion of propionate.

**Author Contributions:** Conceptualization, M.G. and J.K.; methodology, M.G. and J.K.; software, J.K.; validation, M.G., R.A. and J.K.; formal analysis, R.A.; investigation, M.I.V., J.M., M.G. and J.K.; resources, M.G.; data curation, M.I.V., J.M., M.G. and J.K.; writing—original draft preparation, M.I.V., J.M. and M.G.; writing—review and editing, R.A. and J.K.; visualization, R.A.; supervision, M.G. and J.K.; project administration, M.G. and J.K.; funding acquisition, M.G. All authors have read and agreed to the published version of the manuscript.

**Funding:** This research was funded by the Faculty of Agricultural and Food Sciences, Vice-Rectory for Research, Development, and Artistic Creation (VIDCA) of Universidad Austral de Chile, and received no external funding.

**Institutional Review Board Statement:** All animal procedures were performed in accordance with the UK Animals (Scientific Procedures) Act and associated guidelines, and approved by the Animal Ethics Committee of the Austral University of Chile (approval number 144/2013).

**Informed Consent Statement:** Not applicable.

**Data Availability Statement:** The data supporting this study's findings are available from the corresponding author upon reasonable request.

**Acknowledgments:** The authors would like to acknowledge staff from the Austral Experimental Research Station, which helped with the care and handling of the animals.

**Conflicts of Interest:** The authors declare no conflict of interest.

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
