# Peer review of "In Vivo Digestibility and In Vitro Fermentation of High Dietary Fiber Forages in Growing Pigs’ Diets"

_fermentation, doi:10.3390/fermentation9050448_

Round 1
Reviewer 1 Report
Observations to manuscript:
The subject of the manuscript submitted is appropriate for the scope of the journal. The aim of the study is attractive since the study provides a sustainable feed strategy for pigs’ production. However, some recommendations have been made to the paper:
1. Throughout the manuscript, several scientific names and Latin expressions should be italicize.
2. Headings in the abstract are not required to be written. I highly recommend to briefly include the experimental design in the abstract.
3. The introduction reads well, but I recommend to summarizes and highlight key research related to the topic to set out the current knowledge.
4. Ethical approval is required for studies involving animals. The protocol code should be mentioned in the material and methods.
5. Materials and Methods are hard to follow and understand, this section should be written in more detail and organization (subdividing the information may help). Techniques should be briefly described. Some analyzes are mentioned twice throughout the materials and methods section (VFA, NH3), the author should not repeat the information.
6. “Wheat middling” is reported as a pure ingredient incubated on line 144, but it is not reported in the results (Tables 3 and 4).
7. Throughout the manuscript, the “p-value” format must be the same.
8. In general, the discussion is hard to follow due to the lack of connection words between sentences. In addition, the results found should be linked to the possible implications of the results found in the in vivo digestibility and in vitro fermentation parameters in growing pigs. What biological effects can be promoted by these changes? How does the animal benefit?
9. I highly recommend including more recent references, since a high percentage of the cited references are more than 8 years old since they were published.
Line 37-39: I suggest to rephrase sentence. What has been evaluated?
Line 43-49: The author could provide some research results related to the effects of summer brassica dietary supplementation to enrich the paragraph; instead of describing the nutritional composition of the crop.
Line 51-53: Please provide at least one more reference to support this sentence.
Line 53-54: Please provide a reference that support this idea.
Line 69: Please provide information about the sample preparation (where and when the plants were collected).
Line 70, 128: The time of the drying process should be mentioned.
Line 71: I highly recommend to restructure the paragraph, introduce the experimental design, animal treatments, and managements.
Line 79: Units of measurement must be separated from the value
Line 86: Mention the chemical analysis and cited the established method.
Line 88: AARS?
Line 97: “Total fecal collection was made during the last 3 days of each 7-day period, avoiding contamination”. This information is already mention above.
Line 114: Sentence is not clear. The author must specify what he is referring to.
Line 130-131: Provided information of the equipment (gas chromatograph and spectrophotometer).
Line 132: Paragraph is hard to understand, this needed to be re-written. The fecal inoculum referred to was used for the in vitro gas technique or for the in vitro post-gastric fermentation test.
Line 165-168: The conditions of the gas chromatograph and the standard used should be mentioned.
Line 169-172: Paragraph is not clear. Please described this technique with sufficient details
Line 214: “3.1. Experiment 2: In vitro Fermentation.” Please place this sentence outside Table 2.
Line 220: Tables 3 and 4 must have the same format for all data values.
Line 246-250: Paragraph is hard to follow. Consider the use of linking words.
Line 261, 263, 269, 284, 299: The author refers to “other studies”, “other authors”, but only one study is cited.
Line 283: Please mention which microbial population the author is referring to.
Line 286: The author could briefly mention the differences between the bacteria found in the large intestine of pigs and ruminants. Please provide a reference.
Line 291: “The results of GP are in accordance with VFAs concentrations, as there is a strong relationship between GP and VFA concentration”, I suggest to discuss more this argument.
Line 297, 316: A reference must be included in this paragraph.
Line 328: Specify that these findings are in vitro.
Author Response
First of all, we would like to thank the reviewer's effort to provide a thorough review.
Resonses to each comment/suggestion are listed below:
Reviewer 1:
The subject of the manuscript submitted is appropriate for the scope of the journal. The aim of the study is attractive since the study provides a sustainable feed strategy for pigs’ production. However, some recommendations have been made to the paper:
- Throughout the manuscript, several scientific names and Latin expressions should be italicize.
A: Amended
- Headings in the abstract are not required to be written. I highly recommend to briefly include the experimental design in the abstract.
A: Headings were deleted. Unfortunately we cannot include experimental designs in the abstract due to the 200 word limit in the author’s instructions.
- The introduction reads well, but I recommend to summarizes and highlight key research related to the topic to set out the current knowledge.
A: included (please see lines 68-70)
- Ethical approval is required for studies involving animals. The protocol code should be mentioned in the material and methods.
A: Ethical approval and protocol code are stated in its corresponding section (please see lines 400-402)
- Materials and Methods are hard to follow and understand, this section should be written in more detail and organization (subdividing the information may help). Techniques should be briefly described. Some analyzes are mentioned twice throughout the materials and methods section (VFA, NH3), the author should not repeat the information.
A: amended
- “Wheat middling” is reported as a pure ingredient incubated on line 144, but it is not reported in the results (Tables 3 and 4).
A: Amended
- Throughout the manuscript, the “p-value” format must be the same.
A: Amended
- In general, the discussion is hard to follow due to the lack of connection words between sentences. In addition, the results found should be linked to the possible implications of the results found in the in vivo digestibility and in vitro fermentation parameters in growing pigs. What biological effects can be promoted by these changes? How does the animal benefit?
A: The discussion was totally rewritten to make it easier to read and follow. We apologize and appreciate your comments. Please read 199-716.
- I highly recommend including more recent references, since a high percentage of the cited references are more than 8 years old since they were published.
A: Amended. We included 4 more references from 2015 onwards. Fiber effects in nutrition and gut health in pigs: Lindberg, 2014; Jha et al., 2019; Jang et al., 2019; Zhao et al., 2020.
Line 37-39: I suggest to rephrase sentence. What has been evaluated?
A: amended
Line 43-49: The author could provide some research results related to the effects of summer brassica dietary supplementation to enrich the paragraph; instead of describing the nutritional composition of the crop.
A: Included. Please see lines 43-46
Line 51-53: Please provide at least one more reference to support this sentence.
Included. Jha et al 2019 was added supporting new findings of fiber addition
Line 53-54: Please provide a reference that support this idea.
Included. We added 3 references for the assumptions of the sentence
Line 69: Please provide information about the sample preparation (where and when the plants were collected).
A: added (lines 71-77)
Line 70, 128: The time of the drying process should be mentioned.
A: added (line 79)
Line 71: I highly recommend to restructure the paragraph, introduce the experimental design, animal treatments, and managements.
A: Many thanks, paragraph was restructured (lines 81 – 99)
Line 79: Units of measurement must be separated from the value
A: Amended
Line 86: Mention the chemical analysis and cited the established method.
A: Added (please see lines 84-90)
Line 88: AARS?
A: deleted
Line 97: “Total fecal collection was made during the last 3 days of each 7-day period, avoiding contamination”. This information is already mention above.
A: both sentences were merged (lines 112 - 113)
Line 114: Sentence is not clear. The author must specify what he is referring to.
A: Amended (Please see line 131)
Line 130-131: Provided information of the equipment (gas chromatograph and spectrophotometer).
A: Included
Line 132: Paragraph is hard to understand, this needed to be re-written. The fecal inoculum referred to was used for the in vitro gas technique or for the in vitro post-gastric fermentation test.
A: Amended (lines 147-155)
Line 165-168: The conditions of the gas chromatograph and the standard used should be mentioned.
A: included (lines 185-187)
Line 169-172: Paragraph is not clear. Please described this technique with sufficient details
A: amended
Line 214: “3.1. Experiment 2: In vitro Fermentation.” Please place this sentence outside Table 2.
A: Many thanks, amended
Line 220: Tables 3 and 4 must have the same format for all data values.
A: Amended
Line 246-250: Paragraph is hard to follow. Consider the use of linking words.
A: amended
Line 261, 263, 269, 284, 299: The author refers to “other studies”, “other authors”, but only one study is cited.
A: amended
Line 283: Please mention which microbial population the author is referring to.
A: included (lines 304-310)
Line 286: The author could briefly mention the differences between the bacteria found in the large intestine of pigs and ruminants. Please provide a reference.
A: included (lines 304-310)
Line 291: “The results of GP are in accordance with VFAs concentrations, as there is a strong relationship between GP and VFA concentration”, I suggest to discuss more this argument.
A: amended (see lines 318-322)
Line 297, 316: A reference must be included in this paragraph.
A: Included
Line 328: Specify that these findings are in vitro.
A: Included
Reviewer 2 Report
Specific comment:
If the in vitro experiment is first in both title and objective, all other sections of the article should begin with the same experiment, and not, as now, with in vivo. In such a case, it would be better to have good consistency between the order of experiments in all sections of the paper.
Regarding the design chosen: it would be better to present arguments for the suitability of the Latin square design in an experiment with growing animals.
The diets used in the in vivo experiment (Table 1) are described as iso-nitrogen (chemically analyzed) and iso-energetic (calculated). But the data shows that they are not well balanced in terms of all energy sources - ash, crude fiber, NDF and ADF, ether extract. Diets are well balanced only in terms of crude protein and dry matter. This may raise serious doubts about the correctness of the experimental design and the results obtained from that experiment.
General questions:
How many samples of the three experimental diets were analyzed chemically? Add information about n on line 86 and in the heading of Table 1.
Regarding the in vivo digestibility experiment: the results obtained showed a significantly higher gross energy and ash digestibility in the experimental feed groups. But precisely these groups have an unbalanced concentration of ash and other energy sources. Are these results independent of the nutrient concentrations shown? Also, is the conclusion drawn in this regard, correct?
L. 95 - For experiment 1 - animal feeding is ad lib, what is the water consumption?
Table 2, after "Apparent fecal digestibility (in vivo) %" should be added that these values are mean × ± SEM or SD?
Table 3 is missing superscripts that reflect the statistical differences of GP72.
In Tables 3 and 4, replace "Sem" with "SEM".
In the Discussion: the abbreviations DF, IF, SF are not introduced. This gap actually makes the discussion difficult to understand.
Author Response
Thanks for your review. All your comments/suggestions were attended and listed below
Specific comment:
If the in vitro experiment is first in both title and objective, all other sections of the article should begin with the same experiment, and not, as now, with in vivo. In such a case, it would be better to have good consistency between the order of experiments in all sections of the paper.
A: We changed the order and now in vivo digestibility it mentions first both in the title and objectives. This order would be better as the in vitro experiment analyses post-gastric fermentation.
Regarding the design chosen: it would be better to present arguments for the suitability of the Latin square design in an experiment with growing animals.
Latin squares are carried out in a short period of time 5 adaptation days, 4 day to adapt to diets and 4 to collect feces according to Adeola (2001). If you search mostly all digestibility trials are carried out in a Latin square design, due to the necessity to limit the number if animals to get the significant differences. Dr. Hans Stein from the University of Illinois is probably the person with the highest number of digestibility trials results published in the journal of animal sciences. All studies use a Latin square. If we need to the look for growth performance results (N=8), then a complete random design is mostly used. To get used to the next diet, each pig has 4 days to adapt, and a colorant (ferrous oxide) is used to start and finish each period.
More information:
- Schneider and Flatt, 1975. The Evaluation of Feeds Through Digestibility Experiments, University of Georgia Press, 423 pp.
The diets used in the in vivo experiment (Table 1) are described as iso-nitrogen (chemically analyzed) and iso-energetic (calculated). But the data shows that they are not well balanced in terms of all energy sources - ash, crude fiber, NDF and ADF, ether extract. Diets are well balanced only in terms of crude protein and dry matter. This may raise serious doubts about the correctness of the experimental design and the results obtained from that experiment.
Diets are almost the same and the ingredients that differe were the fibrous source. In terms of CP, the three diets were very similar (20.93, 20.91, 20.58% showing a differences of 1.6% among them; iso is less than 5% of differences). Gross energy and net energy are almost the same. When varying the fibrous ingredients, the proportions of soluble and insoluble fiber content vary, unfortunately soluble fiber is not reported due to the difficulties to analyze it. It is difficult to balance diet in terms of all energy sources, but this is not the aim of the study, which is to evaluate new include fibrous ingredients in pig diets. We added a statement in the discussion.
General questions:
How many samples of the three experimental diets were analyzed chemically? Add information about n on line 86 and in the heading of Table 1.
A: One feed sample was taken per week. We pooled this samples for chemical analyses and in vitro fermentation trial.
Regarding the in vivo digestibility experiment: the results obtained showed a significantly higher gross energy and ash digestibility in the experimental feed groups. But precisely these groups have an unbalanced concentration of ash and other energy sources. Are these results independent of the nutrient concentrations shown? Also, is the conclusion drawn in this regard, correct?
A: The differences are a consequence of the nature between the fibrous ingredients. We do not relate the results obtained here to the ashes, as turnip inclusion contained the greater ash concentration (>1%), resulted in lower ash digestibility compared with fodder rape. In this regard GE digestibility of turnip should have been lower if the effect was due to differences in ash concentration and digestibility.
- 95 - For experiment 1 - animal feeding is ad lib, what is the water consumption?
A: Pigs had ad libitum access to water (line 95)
Table 2, after "Apparent fecal digestibility (in vivo) %" should be added that these values are mean × ± SEM or SD?
A: included
Table 3 is missing superscripts that reflect the statistical differences of GP72.
A: included
In Tables 3 and 4, replace "Sem" with "SEM".
A: Amended
In the Discussion: the abbreviations DF, IF, SF are not introduced. This gap actually makes the discussion difficult to understand.
A: This abbreviation were defined in the introduction (Please see L39, L41 and L42)
Round 2
Reviewer 1 Report
Comments were addressed in this revised version.